# Light Regulation of LoCOP1 and Its Role in Floral Scent Biosynthesis in *Lilium* ‘Siberia’

**DOI:** 10.3390/plants12102004

**Published:** 2023-05-16

**Authors:** Yang Liu, Qin Wang, Farhat Abbas, Yiwei Zhou, Jingjuan He, Yanping Fan, Rangcai Yu

**Affiliations:** 1The Research Center for Ornamental Plants, College of Forestry and Landscape Architecture, South China Agricultural University, Guangzhou 510642, China; liuyang@szpt.edu.cn (Y.L.); wq2638341289@stu.scau.edu.cn (Q.W.); farhatmerani@yahoo.com (F.A.); zhouyiwei6333@163.com (Y.Z.); hejingjuan@stu.scau.edu.cn (J.H.); 2Guangdong Key Laboratory for Innovative Development and Utilization of Forest Plant Germplasm, South China Agricultural University, Guangzhou 510642, China; 3College of Life Sciences, South China Agricultural University, Guangzhou 510642, China

**Keywords:** *Lilium* ‘Siberia’, floral scent, light, LoCOP1, LoMYBs

## Abstract

Light is an important environmental signal that governs plant growth, development, and metabolism. Constitutive photomorphogenic 1 (COP1) is a light signaling component that plays a vital role in plant light responses. We isolated the *COP1* gene (*LoCOP1*) from the petals of *Lilium* ‘Siberia’ and investigated its function. The LoCOP1 protein was found to be the most similar to *Apostasia shenzhenica* COP1. LoCOP1 was found to be an important factor located in the nucleus and played a negative regulatory role in floral scent production and emission using the virus-induced gene silencing (VIGS) approach. The yeast two-hybrid, β-galactosidase, and bimolecular fluorescence complementation (BiFC) assays revealed that LoCOP1 interacts with LoMYB1 and LoMYB3. Furthermore, light modified both the subcellular distribution of LoCOP1 and its interactions with LoMYB1 and MYB3 in onion cells. The findings highlighted an important regulatory mechanism in the light signaling system that governs scent emission in *Lilium* ‘Siberia’ by the ubiquitination and degradation of transcription factors via the proteasome pathway.

## 1. Introduction

*Lilium* ‘Siberia’ (Oriental hybrid) is a perennial flowering plant that is well known in the global flower market for its large, snow-white, and fragrant flower tissue, making it an excellent resource for the research of floral scent. The floral scent emission from *Lilium* ‘Siberia’ usually has a diurnal rhythm that coincides with pollinator activity, and the sepals and petals are the principal sources of this emission [1,2,3,4]. Terpenoids and benzenoids like linalool, ocimene, myrcene, methyl benzoate, and ethyl benzoate are main volatile compounds in the flower fragrance of *Lilium* ‘Siberia’ [4,5,6,7,8,9]. Floral volatiles are volatile organic compounds (VOCs) released by plants, and flowers are the primary source of flower scent synthesis and emission mechanisms; however, other floral tissues such as anther, stigma, and leaves also play important roles in the aforementioned mechanism [4,10,11]. The abundances and variations in floral VOCs vary greatly among flowering plants [12]. Floral volatiles are categorized into four categories based on their origin, biosynthesis, and function: terpenoids, phenylpropanoids, fatty acid derivatives, and amino acids [13,14,15,16,17]. Various environmental and endogenous factors influence the synthesis, composition, and emission of floral scents [18].

Numerous investigations have shown that external environmental factors, including temperature [19,20,21], light [20], and atmospheric CO_2_ concentration [22], and endogenous factors, such as circadian rhythms [23,24,25], floral developmental stages [26,27], and plant age [28], are accountable for the biosynthesis and emission mechanism of floral scent. Previous studies showed that terpenoids and indole emissions in maize plants increase with temperature, peaking between 22 °C and 27 °C and thereafter declining [29]. Light intensity is one of the environmental elements that has a significant impact on floral fragrance emission. The overall amount and the number of volatile compounds released by various plants have been shown to increase when light intensity increases [30]. Higher light intensity and temperature can cause the emission of floral aroma from *Lilium* ‘Siberia’, demonstrating that light intensity and temperature have a substantial impact on the number and amount of floral scent emission [20]. Under continuous irradiation, the levels of (E)-βocimene and myrcene released in *Antirrhinum majus* were higher than in the dark [14,31,32]. The volume and variety of volatile organic compounds emitted by *Lilium* ‘Siberia’ during the course of a single day followed the pattern of initially increasing and then decreasing as the day progressed [33].

Photoreceptors regulate the activities of constitutive photomorphogenic 1 (COP1), a key regulator of light signaling, to perform a large portion of their signaling function. COP1 is found in many eukaryotes, and it is well conserved in plants and animals. It acts as a RING E3 ubiquitin ligase, promoting the proteasome-mediated degradation of a variety of substrates [34]. The COP1/SPA unit inhibits light signaling in the dark, primarily by ubiquitination, and thereby causes the degradation of photomorphogenesis positive regulators, which are predominantly transcription factors. Photoreceptors inhibit COP1/SPA activity when plants are exposed to light, resulting in the stability of COP1/SPA substrates, which promotes photomorphogenesis [35]. Phytochrome interacting factors (PIFs), which are antagonistic integrators of light signaling, are similarly stabilized by the COP1/SPA complex [36]; however, the underlying mechanism in *Lilium* ‘Siberia’ is completely unknown. Only *Arabidopsis*, rice, and apples have been studied extensively in terms of COP1. Previous studies showed that the floral fragrance of *Lilium* ‘Siberia’ is influenced by light [5,20], and COP1 is a key light signaling component involved in plant light responses. However, it is unclear whether LoCOP1 affects the floral fragrance of *Lillium* ‘Siberia’ in response to light variations. To acquire further about LoCOP1, we isolated and functionally characterized the *LoCOP1* gene from *Lilium* ‘Siberia’.

## 2. Results

### 2.1. Bioinformatics Analysis of LoCOP1 in Lilium ‘Siberia’

A blast search of the lily genome database yielded one *COP1*-like gene based on the sequence of *Arabidopsis COP1*. The genomic sequence obtained was used to design primers, and the corresponding full-length cDNA clone was generated using a reverse transcription polymerase chain reaction (RT-PCR) (Appendix A). The full-length complementary DNA (cDNA) sequence of *LoCOP1* contains an open reading frame (ORF) 2076 bp (Appendix A), encoding polypeptides of 691 amino acids. The deduced amino-acid sequence of LoCOP1 was aligned with that of monocotyledonous plants such as ZmCOP1, SiCOP1, and OsCOP1. LoCOP1 exhibited strong sequence identity in the N-terminal zinc-RING domain and C-terminal WD-40 region. This polypeptide contains DNA zinc finger domains and WD-40 repeat sequences that are highly conserved (Figure 1A).

To examine the evolutionary relationships between LoCOP1 and COP1s from other plants, a neighbor-joining phylogenetic tree was built using the amino-acid sequences of LoCOP1 and other homologs from 24 plant species (Figure 1B). LoCOP1 was shown to be the most closely related to *Apostasia shenzhenica* (PKA52651.1) in this study since they were both clustered in the same clade. LoCOP1 was found in mostly monocotyledonous plants and was clustered into one clade, while COP1s from dicotyledons plants were found in another.

### 2.2. Tissue-Specific and Expression Pattern Analysis of LoCOP1 in Lilium ‘Siberia’ 

Real-time quantitative PCR using gene-specific primers was used to determine if *LoCOP1* transcript levels are tissue specific and impacted by plant development. LoCOP1 was found to be highly expressed in the sepal, pistils, and the lowest amounts in the root (Figure 2A). The expression of *LoCOP1* in the sepal gradually increased and then decreased as the flowers opened, but there was no significant change in the expression of *LoCOP1* in the petal (Figure 2B).

### 2.3. Subcellular Localization of LoCOP1 Is Controlled by Light

*LoCOP1* full-length sequences were evaluated utilizing several web servers. According to the bioinformatics analysis, LoCOP1 was found in the nucleus. The sequences were introduced into a p35 S-EGFP-1 vector, and onion epidermis cells were used to confirm the predicted results. When LoCOP1-GFP was expressed in onion epidermal cells, it produced green speckles in the nucleus and blue speckles under DAPI illumination (Figure 3A), showing that LoCOP1 is located in the nucleus. 

Light is an essential environmental factor for metabolism, and the subcellular distribution of COP1 changes depending on the light condition. We tested the hypothesis that light levels affect LoCOP1 distribution by transiently expressing a LoCOP1-GFP fusion protein in onion epidermal cells for 48 h in either a light or dark environment. Green fluorescence could be seen with the various protein combinations in both conditions, although more green speckles were visible in the light (Figure 3B). Furthermore, the quantity of fluorescent nuclei under different conditions varies greatly (Figure 3C), indicating that LoCOP1 can translocate in a light-dependent way.

### 2.4. LoCOP1 Negatively Regulate the Synthesis of Fragrance Emission in Lilium ‘Siberia’

*LoCOP1* expression was inhibited and transiently overexpressed in *Lilium* ‘Siberia’ flowers to assess its function in floral scent. The result showed that after the transient overexpression of *LoCOP1*, the expression level of *LoCOP1* was increased (Figure 4A), and the emissions of ocimene, linalool, ethyl benzoate, and methyl benzoate declined by 60.4%, 58.8%, 76.4%, and 89.3%, respectively (Figure 4B). In the meantime, *LoCOP1* transcript levels were significantly reduced following VIGS-mediated silencing compared to unsilenced control flowers (Figure 4C), leading to a higher concentration of floral volatiles. The released amount of ocimene, linalool, ethyl benzoate, and methyl benzoate were increased by 2.8, 0.9, 0.4, and 2.1 times, respectively (Figure 4D). The overexpression of *LoCOP1* inhibited the emission of fragrance, and the suppression of *LoCOP1* promoted the emission of floral aroma, indicating that LoCOP1 played a negative regulatory role in the synthesis of floral scent emission in *Lilium* ‘Siberia’.

### 2.5. LoCOP1 Interacts with LoMYBs in Yeast and Onion Cells

In *Arabidopsis*, AtCOP1 interacts directly with the MYB transcription factors PAP1 and PAP2 to regulate anthocyanin accumulation [37]. Similarly, in apples, MdCOP1 interacts with MdMYB1 to regulate light-induced anthocyanin production and red fruit color [38]. We used a yeast two-hybrid assay to evaluate whether LoCOP1 regulates the key floral scent related to LoMYB1 and LoMYB3 in lily. LoMYB1 and LoMYB3 also showed tissue-specific expression patterns and potentially play crucial roles in floral aroma production in *Lilium* ‘Siberia’ (Appendix A). The Y2H assay showed that LoCOP1 interacted with LoMYB1 and LoMYB3 (Figure 5A). The results of the β-galactosidase experiments also indicated that LoCOP1 interacts with LoMYB1 and LoMYB3 (Figure 5B). To verify the aforementioned findings, the interaction between LoCOP1 and the LoMYBs was assessed using a BiFC (bimolecular fluorescence complementation) assay. We fused LoCOP1 to the C-terminal fragment of GFP (GFP^C^) and LoMYB1 and LoMYB3 to the N-terminal fragment of GFP (GFP^N^), and both fusion proteins were transiently introduced into onion epidermal cells. GFP fluorescence was largely found in the nucleus of cells cotransformed with two combinations: LoCOP1-GFP plus LoMYB1-nGFP^N^ and LoCOP1-GFP^C^ plus LoMYB3-GFP^N^ (Figure 5C). In contrast, no fluorescence was seen in the control combinations of the empty pSPYNE-35S vector plus LoCOP1-GFP^C^ or the empty pSPYCE-35S vector with either LoMYB1-GFP^N^ or LoMYB3-GFP^N^. These findings indicate that LoCOP1 can interact with LoMYB1 and LoMYB3 in plant cells in vivo. 

### 2.6. LoCOP1 Interactions with LoMYBs in Onion Cells Were Controlled by Light

Given that LoCOP1 localization in onion cells is light dependent, we investigated whether the interaction of LoCOP1 and LoMYBs in a BiFC (bimolecular fluorescence complementation) system was light dependent. The cotransformation of LoCOP1-GFP^C^ plus LoMYB1-GFP^N^ and LoCOP1-GFP^C^ plus LoMYB3-GFP^N^ in onion epidermal cells produced strong GFP fluorescence in nuclear speckles in both darkness and white light (Figure 6A,B), but more fluorescence was observed in the continue dark than in the light condition (Figure 6C), indicating that LoCOP1 interactions with LoMYB1 and LoMYB3 in onion epidermal cells were controlled by light. 

## 3. Discussion

The composition, synthesis, and emission of floral fragrances are influenced by light [39,40,41,42]. In this study, the results show that the emissions of linalool, ocimene, allo-ocimene, and ethyl benzoate in flowers treated with continuous light were significantly increased compared to those of flowers treated with continuous darkness (Appendix A), and their concentrations were upregulated when the culture conditions from dark turned to light. Contrarily, these volatile compounds were downregulated when the culture conditions from light turned to dark (Appendix A). These results suggested that light might affect synthesis and the emission of floral scent.

Accumulated evidence has indicated that COP1 acts as a central switch of light signaling networks in *Arabidopsis*. It mediates light responses by integrating signals from various photoreceptors and regulating a batch of downstream factors [43,44,45]. The majority of what is now known about the function of the COP1 family in higher plants comes from research on *Arabidopsis*, which has four COP1-like genes. We obtained a *COP1* cDNA clone (*LoCOP1*) from the petals of *Lilium* ‘Siberia’ flowers in this study. LoCOP1’s deduced amino-acid sequence includes a RING-finger domain in the C-terminus, a coiled–coil region in the middle, and seven WD-40 repeats in the N-terminus. These three domains are the key functional areas that have been found to be highly conserved in other COP1s from other plants. COP1 acts as a central switch in light signal transduction in *Arabidopsis* by physically interacting with upstream light receptors via its N-terminal WD-40 repeat domain and ubiquitinating itself or other downstream transcription factors (TFs) via its C-terminal RING-finger, which is commonly conserved in a subclass of ubiquitin protein ligases [43,44,45]. Because LoCOP1 and AtCOP1 have similar sequences and function domains, LoCOP1 may be a counterpart of AtCOP1. According to the results of the sequence alignment, LoCOP1 is highly homologous to other COP1s. In our phylogenetic analysis, we discovered a clear contrast between dicotyledonous and monocotyledonous plants. The highly comparable sequences of LoCOP1 suggest that COP1 function may be retained in the majority of plant species.

Protein subcellular localization is critical for its function. Since LoCOP1 is positioned in the nucleus, it is advantageous to offer favorable space conditions for access to nuclear genes and photoreceptors, which transfer perceived light signals via interacting proteins and other signaling components. As a result, the nuclear localization of photoreceptors and some signaling components suggests that at least one branch of light signaling takes place within the nucleus [46,47]. Under light conditions, green fluorescence within the nucleus and the number of speckles were weaker, showing that the light-induced LoCOP1 protein diffuses from the nucleus to the cytoplasm. LoCOP1 and AtCOP1 can both translocate in a light-dependent way. COP1 acts as a light-responsive nucleocytoplasmic-shuttling parent in *Arabidopsis* and is nuclear-localized in the dark, and its abundance is lowered by light, according to previous studies [48]. This corresponded not only with the increase in MdCOP1 abundance in the cell nucleus caused by the dark treatment in apples but also with the increase in COP1 content in the nucleus of ‘Jingxiu’ grapes grown in the shade [38,49]. The differential nuclear abundance of COP1 under different light conditions is primarily assessed by phytochrome and blue light receptors. Because of the different subcellular locations of PHYA and PHYB, which regulate the levels of COP1 in nuclei in different ways, COP1 that is transported from cytoplasm to the nucleus may be impacted by the inhibition of phytochrome PHYA, which is localized in the cytoplasm. However, PHYB, which is present in the nucleus under white light and in the cytoplasm under black light, is principally responsible for COP1 protein breakdown within the cell nucleus and the transit from the nucleus to the cytoplasm [50]. Furthermore, when rice is exposed to blue light, the blue light receptor OsCRY1b interacts with OsCOP1, resulting in a decrease in the COP1 concentration in the nucleus. Aside from photoreceptors, various regulators are involved in the process of light-adjusting COP1 protein in nuclear mass distribution.

A pleiotropic gene COP/DET/FUS expression product is a composite that functions as a signal in a signal transduction system. Under dark conditions, the activity of the COP/DET/FUS protein complex increased, resulting in COP1 protein localization in the nucleus. However, under light conditions, the optical signal can lower COP/DET/FUS complex activity, resulting in COP1 protein localization in the cytoplasm. As a result, COP1 protein accumulation within the nuclear process necessitates the involvement of a multi-effect type COP/DET/FUS expression product, which acts as a regulator of COP1 protein localization. Further research has revealed that, in the dark, the COP/DET/FUS combination can efficiently stimulate the interaction of COP1 and a novel HYH bZIP protein, resulting in HYH protein breakdown [51]. It was also discovered that MID (an integral component of topoisomerase) exists in *Arabidopsis* plants, and that it can interact with COP1 to alter COP1 accumulation in the nucleus under dark conditions [52]. Furthermore, the structure of the COP1 protein is important for its own subcellular distribution. The separate domains comprising the nuclear localization signal (NLS) and the cytoplasmic localization signal (CLS) were combined to recreate the light regulation of nuclear localization [53]. When exposed to light, importin recognizes and binds to the nucleus signal NLS in the cytoplasm, regulating the process of COP1 transport from the cytoplasm into the nucleus. Distinct importins containing different nucleoproteins have been discovered in rice, with 1a, 1b, and 2 having precedence to mediate the localization of the COP1 protein [54]. Nonetheless, nuclear exclusion was mediated by CLS, a unique and distinct signal that borders the zinc-finger and coiled–coil motifs and has the ability to reroute a heterologous nuclear protein to the cytoplasm. However, how can COP1 be controlled? CSU1 (constitutively photomorphogenic 1 suppressor1) is a RING-finger E3 ubiquitin ligase that plays an important role in COP1 homeostasis by ubiquitinating and degrading COP1 in dark-grown seedlings [55]. 

In taller plants, COP1 plays the role of a molecular switch for light-induced plant development processes, such as photomorphogenesis [51,56,57,58], flowering time [59], photoperiodic growth [60], stomatal development [61], and anthocyanin accumulation [62,63]; however, there is no report about the floral fragrance. The tissue-specific expression study revealed that *LoCOP1* gene expression was found in all tissues and expressed at relatively high levels in sepals, indicating that LoCOP1 may have a function in the floral scent production pathway of *Lilium* ‘Siberia.’ Significantly, the high expression level of *LoCOP1* in the sepals and petals of *Lilium* ‘Siberia’ allowed us to investigate its effect on volatile production in flowers that produce an array of volatile compounds originating from diverse biochemical origins—sesquiterpenes, monoterpenes, fatty acid derivatives, and phenylpropanoid volatile compounds. When *LoCOP1* was overexpressed in transiently transformed plants, it resulted in the downregulation of numerous genes from the shikimate pathway and downstream scent-related genes from the phenylpropanoid pathway (i.e., LoAAT and *LoTPSs*), resulting in significantly lower levels of numerous floral volatiles (e.g., ocimene, linalool, ethyl benzoate, methyl benzoate, and others). A contrast trend was observed in flowers with reduced LoCOP1. LoCOP1 plays a negative regulatory role in the synthesis of fragrance emission, as evidenced by its effect on flower scent. *Cop1* mutants in *Arabidopsis* produced more anthocyanin and appeared much redder in color than the wild type and still had anthocyanin synthesis even when treated with completely dark [64,65]. The *cop1* mutant’s pleiotropic phenotypes suggest that COP1 as an inhibitor regulates not only photomorphogenesis but also other developmental processes [63,66,67]. This conclusion supports prior findings that MdCOP1s are negative regulators of apple fruit coloration [38]. Similarly, in ‘Jingxiu’ grapes grown in shadows, the nuclei COP1 levels rose, lowering anthocyanin synthesis and making the fruit color lighter [68,69]. Given the similar function of COP1 in different plants, it is plausible to hypothesize that LoCOP1 in *Lilium* ‘Siberia’ has E3 ubiquitin ligase activity and the ability to sense light signals by physically engaging with upstream light receptors to regulate the downstream protein. In plants, how does the COP1 protein function? In the dark, nuclear COP1 inhibits photomorphogenesis by targeting a subset of transcription activators, including the basic helix-loop-helix transcription factor HFR1 [70], the basic zipper transcription factors HY5 and HY5 homolog [51,56], and the B-box-type zinc-finger transcription factors CO and LIGHT-REGULATED A. The light-controlled nuclear depletion of the COP1 protein, on the other hand, allows for the accumulation of TFs that activate target genes. We studied COP1’s role by finding genes that are controlled by it. MYB proteins are another type of crucial TFs controlled by the COP1 protein. Several MYBs have been identified as COP1 targets in *Arabidopsis*. AtMYB21 was a flower-specific transcription factor and was regulated by COP1. COP1 is required to repress the *AtMYB21* gene in seedlings in order to govern the plant’s developmental process [71]. LAF1, another MYB transcription activator involved in the transfer of PhyA signals to downstream responses, similarly serves as a COP1 substrate for degradation [57]. Similarly, BIT1, a MYB transcription factor, plays a key function in modulating blue-light responses. COP1 interacts with BIT1 and regulates its degradation in the dark, whereas CRY1 stabilizes BIT1 in a blue-light-dependent way [58]. MYB transcription factors PAP1/2, which are targets of AtCOP1/SPA, govern the protein stability of PAP1 and PAP2, which are implicated in anthocyanin accumulation [37]. According to previous research, MdCOP1s affect apple fruit coloration by negatively regulating the quantity of the MdMYB1 protein, which is a positive regulator of anthocyanin production and fruit coloration [38,49]. MYB TFs have both been linked to the regulation of floral smell production in the model plant Petunia. Additionally, in *Lilium* ‘Siberia’, floral-specific transcription factors LoMYB1, LoMYB2, and LoMYB3 have been identified as essential regulatory genes for Phenylpropanoid metabolic pathways producing fragrance compounds, despite the fact that they directly bind to the promoters of Phenylpropanoid biosynthesis genes (*LoAAT*). In general, light stimulates the production of anthocyanin-associated MYB genes, which promotes the accumulation of red pigments in plant organs [72,73,74,75]. As a result, we need to investigate whether LoCOP1 inhibits scent emission via regulating the LoMYBs protein. We discovered that LoCOP1 and LoMYBs can interact in both yeast and onion epidermal cells, showing that LoMYBs are tagged LoCOP1 target proteins. As previously stated, light affects LoCOP1’s ability to attract LoMYBs, whereas β- galactosidase activity was not affected by light conditions, implying that light does not affect its binding force between them, because light regulates the amount of LoCOP1 in the nucleus to affect the interaction with LoMYBs. Light signals monitor the ubiquitination activity of the COP1 protein by modifying its interaction with light receptors and modulating its subcellular location from cytoplasm to nucleus because COP1 partitioning between nucleus and cytoplasm has been implicated in the regulation of COP1 activity. Furthermore, we cloned a gene called SUPPRESSOR OF PHYA-105 (SPA) (Appendix A). COP1 and the SPA protein family are both WD-repeat proteins that interact with one another via their coiled–coil domains. COP1 has a RING finger domain, while SPA proteins have a kinase domain. Previously published cop1 and spa1 spa2 spa3 spa4 quadruple mutants show characteristics of light-grown seedlings in complete darkness, indicating that both COP1 and SPA proteins are necessary for COP1/SPA complex activity [76,77,78] When photoreceptors suppress COP1/SPA function, the targets of this ubiquitin ligase accumulate to initiate light responses. Light inhibits COP1/SPA by directly interacting with cryptochromes and SPA1 [45,79,80]. Light exposure also destabilizes SPA1 and SPA2 [81]. In the presence of far-red, red, and blue light, PHY and CRY receptors interact with SPA, causing the breakdown of the COP1-SPA complex and decreasing COP1-SPA interaction [82,83,84]. The yeast results showed that LoCOP1 can interact with LoSPA, a ubiquitin ligase complex that can stop photomorphogenesis. Additionally, LoSPA interacts with LoMYBs in yeast and onion cells in a similar manner to LoCOP1 (Appendix A). Thus, LoSPA may be a LoCOP1 partner for fragrance in the dark.

However, it is unknown whether and how phenylpropanoid-associated MYB TFs are controlled at the posttranslational level in *Lilium* ‘Siberia’ under varied light conditions. AtCOP1/SPA control the protein stability of PAP1 and PAP2 involved in anthocyanin accumulation, demonstrating no change in PAP1 and PAP2 transcript levels, which suggests that the stabilization of PAP1 and PAP2 proteins is sufficient to allow an increase in anthocyanin levels in *Arabidopsis* [37,85]. The overexpression or inhibition of MdCOP1s in apples had little effect on MdMYB1 gene expression but had a large impact on MdMYB1 protein level. MdCOP1s posttranslationally ubiquitinate the MdMYB1 protein to control its breakdown via the 26S proteasome pathway under darkness [38]. The mechanism by which COP1s modify the light-regulated stability of MYB TFs appears to be retained in higher plants.

## 4. Materials and Methods

### 4.1. Plant Material and Growth Conditions

Plants of *Lilium* ‘Siberia’ were cultivated in a growth chamber at 24 °C, relative humidity of 65–80%, and a light/dark cycle of 14/10 h. Light intensity was 8000 lux, and light source was LED (light-emitting diode). To investigate the effect of light on the release of fragrance substances in *Lilium* ‘Siberia’, the flowers were kept in environmental conditions as described above until they had reached the full bloom stage. Then, flowers were treated for 48 h under constant light and continuous dark, respectively. The controlled flowers were kept in a growth chamber under 12 h light/12 h dark cycle. After every 4 h, floral aroma was collected and identified using headspace solid-phase microextraction gas chromatography-mass spectrometry (HS-SPME-GC-MS). To evaluate if the emission of floral aroma is stimulated by light and inhibited by darkness, flowers at the bud stage were treated in the dark and under continuous light until full bloom, Then, the lighting conditions were switched.

### 4.2. Bioinformatics Analysis and Cloning of LoCOP1 

GO term analysis was used to identify LoCOP1 from the *Lilium* ‘Siberia’ transcriptome data as explained previously [1,3]. To determine *LoCOP1* orthologues in other species, the peptide sequence of *LoCOP1* was blasted in the NCBI p blast software using the default parameters. DNAMAN software was used to perform multiple sequence alignment of LoCOP1 with other COP1 obtained from other species. The phylogenetic tree was constructed in MEGA 7 [86]. The amino acid sequence alignment was performed using ClustalW, and the tree was constructed using the NJ method. The bootstrap values were set to 1000. The amino-acid sequences used for multiple sequence alignment and phylogenetic analysis were obtained from the National Center for Biotechnology Information (NCBI). AtCOP1 protein sequence obtained from TAIR database. LoCOP1 open reading frame was cloned into vector pMD19-T (TaKaRa, Otsu, Japan) according to the manufacturer guidelines and validated by sequencing as explained previously [3,87]. Appendix A lists the forward and reverse primers.

### 4.3. RNA Extraction and Quantitative RT-PCR Analysis 

Total RNA was isolated from several tissues (root, bulb, stem, leaf, sepal, petal, stigma, anther, and filament) and flower development phases (bud, open, half-open, blooming, and fade) using plant RNA mini kit (Magen, Guangzhou, China) following the manufacturer’s suggestions. Using the PrimeScript^TM^ First Strand cDNA Synthesis Kit, the RNA was reverse transcribed (Takara, Otsu, Japan). Agarose gel electrophoresis and melting curve analysis were used to confirm the specificity of each primer pair. RT-qPCR was carried out in a total volume of 20 µL utilizing an ABI 7500 Real-Time PCR System, and using GAPDH as an internal standard, as previously described [1]. 

### 4.4. Virus-Induced Gene Silencing of LoCOP1

To confirm the functional analysis of *LoCOP1*, the expression of *LoCOP1* was suppressed in *Lilium* ‘Siberia’ using the barley stripe mosaic virus (BSMV) system. pCaBS-α, pCaBS-β and pCaBSγ are essential components of the BSMV system, they ensure the high-efficiency infectivity and transformation of the virus in the cell [88]. The BSMV-VIGS vectors were kindly provided by Professor Dawei Li (China Agricultural University). We employed BSMV to infect the sepals of *Lilium* ‘Siberia’, and there was no significant difference in sepal growth and development when compared to controls (Appendix A). A 300-bp amplicon of *LoCOP1* was inserted in pCaBSγ empty vector to generate the pCaBSγ: *LoCOP1* constructs to silence *LoCOP1*. Following that, the constructions were transformed into *Agrobacterium tumefaciens* strain EHA105. The solution was injected to the flowers via immersion in the bacterial suspension via vacuum infiltration and kept at 16 °C for 4–5 days. At full bloom, floral volatile compounds were collected and analyzed using GC-MS, as previously described [1]. The experiment used 3–5 biological duplicates.

### 4.5. Overexpression of LoCOP1 in Lilium ‘Siberia’ Flowers

The full-length ORF sequence of *LoOP1* was amplified and cloned into the pOx vector (provided by State Key Laboratory for Conservation and Utilization of Subtropical AGRO-Bioresources, South China Agricultural University, Guangzhou 510642, China) under the control of the ubi promoter to generate pOx-*LoCOP1*. *Agrobacterium* GV3101 was genetically transformed with 200 ng of plasmid DNA (in 100 µL) of empty vector (pOx) and pOx-*LoCOP1*. *Agrobacterium* cells bearing *LoCOP1* were grown overnight at 28 °C in fresh liquid medium containing kanamycin and 25 µg/mL rifampicin. Bacteria were harvested and resuspended at an OD_600_ of 0.6 in MS liquid medium with 100 µM AS (Acetosyringone). Buds of *Lilium* ‘Siberia’ were injected with the bacterial solution. Construction of overexpression vector and *A*. *tumefaciens* infiltration was carried out as previously disclosed [27,89]. After injection, the flower buds were placed in a normal light incubator to continue culturing until full bloom. Flowers in bloom were sampled in order to collect and analyze floral volatiles.

### 4.6. Yeast Two-Hybrid Assays

Yeast two-hybrid assays were carried out using the Matchmaker GAL4-based two-hybrid system, as previously described (Clontech, Palo Alto, CA, USA). Full-length LoCOP1 and LoMYB1, LoMYB2, and LoMYB3 cDNA fragments were introduced into pGADT7 and pGBKT9 (Clontech, Palo Alto, CA, USA). All constructs were transformed into the yeast strain AH109 using the lithium acetate technique, and yeast cells were cultured on minimal medium/-Leu/-Trp according to the manufacturer guidelines (Clontech, Palo Alto, CA, USA). To screen for potential interactions, transformed colonies were plated onto minimal medium/-Leu/-Trp/-His/-Ade and stained with X-gal.

### 4.7. Bimolecular Fluorescence Complementation (BiFC)

The full-length LoCOP1 and LoMYB1 and LoMYB3 sequences were inserted independently into PUC-SPYNE and PUC-SPYCE vectors (with GFP green fluorescent label) to create LoCOP1-YCE and LoMYB1/3-YNE, respectively. The empty vector (control) and recombinant plasmids were transformed into EHA105 competent cells, and various combinations of *Agrobacterium* cultures were co-infiltrated into *N. benthamiana* leaves. Using the *Agrobacterium* infiltration approach, onion epidermal cells were transiently transformed with various combinations of these constructs. Following infiltration, onion epidermal cells were placed in 1/2 MS solid medium and separated into two groups to investigate the effect of continuous light and dark. The infiltrated cells were observed using a Leica DM RXA2 upright fluorescent microscope, as previously described [27,90].

### 4.8. Subcellular Localization 

The full-length fragments, omitting the stop codon, were fused with the GFP gene in the pBIN expression vector, and the resulting vectors were genetically transformed into onion epidermal cells in the same way that BiFC was. 

### 4.9. Headspace Collection and GC-MS Analysis

The headspace collection and GC-MS analysis was performed as described previously [91,92]. The single blooming flower was placed in a 2 L glass bottle with 1.728 µg ethyl caprate added as an internal standard. After 15 min of volatiles equilibrium, a Supelco polydimethylsiloxane (PDMS, containing 50/30 m divinylbenzene/Carboxen) fiber was placed into the bottle to absorb volatiles for 15 min. The trapped flower fragrance components were then examined using a GC-MS system equipped with an Agilent 7890A GC and an Agilent 5975C MSD. By comparing the mass spectra and retention periods to legitimate standards, the volatiles were identified. Peak regions and the internal standard were used for quantification.

## 5. Conclusions

With a daily cycle of highs and lows, the volatile fragrant of *Lilium* ‘Siberia’ has a clear effect on its fragrance production: light can increase fragrance release, but darkness suppresses the fragrance release. This was discovered in a *Lilium* ‘Siberia’ petal. It was observed that LoCOP1 was an essential factor located in the nucleus, which performed a negative regulatory role in fragrance synthesis and emission. Both yeast and onion epidermal cells showed the interaction of LoMYB1 and LoMYB3 with LoCOP1, showing that LoCOP1 is a homolog of AtCOP1 and that LoMYB1 and LoMYB3 are target proteins of LoCOP1. Our results show that LoCOP1 may ubiquitinate LoMYBs to control its degradation via the 26S proteasome route in the dark.

## Figures and Tables

**Figure 1 plants-12-02004-f001:**
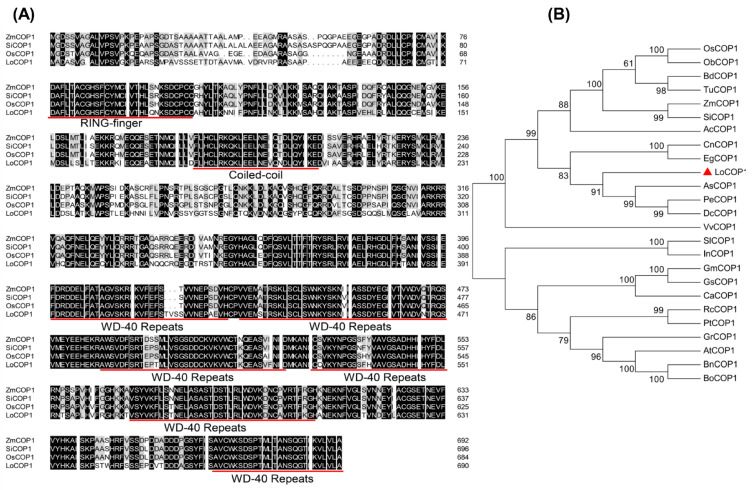
Multiple sequence alignment and phylogenetic analysis of LoCOP1 with other plants. (**A**) Multiple sequence alignment. The conserved zinc ring domain, coiled–coil region, and WD-40 repeat sequences were underlined with red. (**B**) Phylogenetic analysis of LoCOP1 together with COP1 proteins from different plants. Red triangle indicate COP1 from *Lilium* ‘Siberia’. Detailed information of COPs from different plants are listed in Appendix A.

**Figure 2 plants-12-02004-f002:**
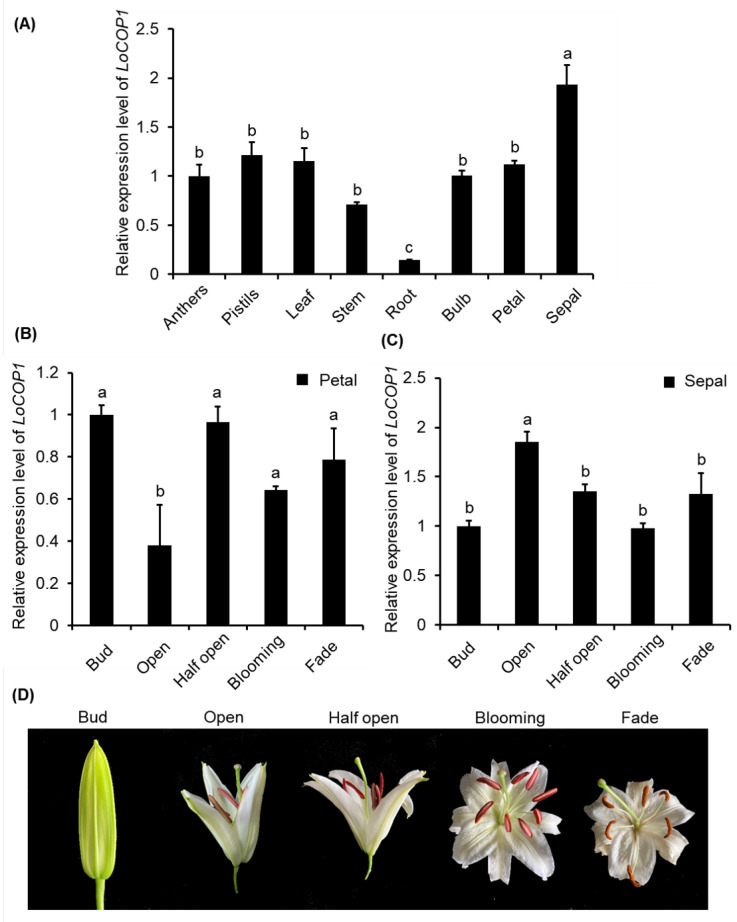
RT-qPCR analysis showing tissue-specific and development-regulated expression of *LoCOP1*. (**A**) Transcript levels of *LoCOP1* in various tissues of *Lilium* ‘Siberia’. (**B**) LoCOP1 transcript levels in the different development stages of petal. (**C**) LoCOP1 transcript levels in the different development stages of sepal. (**D**) Pictorial view of different development stages in floral development of *Lilium* ‘Siberia’. Small letter on bars indicates statistically significant difference at *p* < 0.05. Error bars show the standard deviations from tree biological replicates.

**Figure 3 plants-12-02004-f003:**
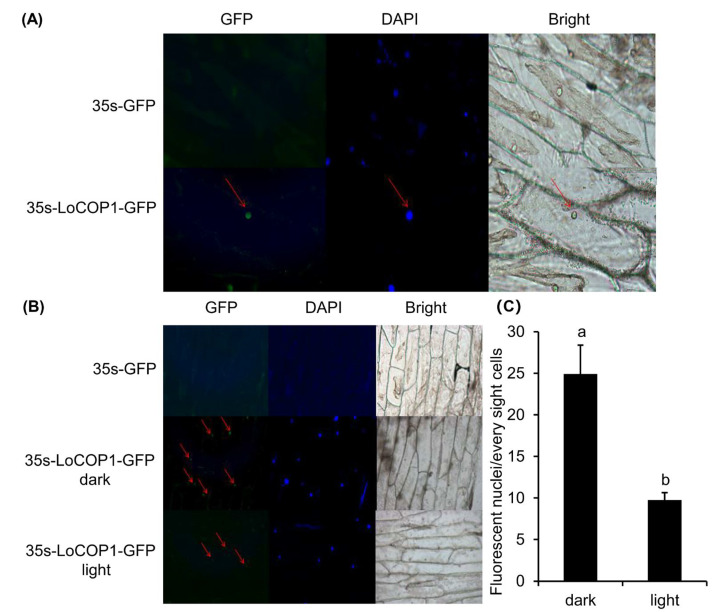
Subcellular localization of LoCOP1 in light-dependent manner. (**A**) Wide-field fluorescence microscopy images of representative cells of Pro35S: GFP-LoCOP1 in onion epidermal cells grown under 16 h light/8 h dark for two days. DAPI staining was used to confirm nuclear localization. (**B**) Confocal microscopy of representative nuclei of Pro35S: GFP-LoCOP1 in onion epidermal grown under white light or dark conditions. Transmission images and chlorophyll fluorescence are also included. (**C**) Number of fluorescent nuclei of the nuclei in seedlings expressing GFP-LoCOP1 grown under white light or dark conditions. Small letter on bars indicate statistically significant difference at *p* < 0.05. Error bars show the standard deviations from tree biological replicates.

**Figure 4 plants-12-02004-f004:**
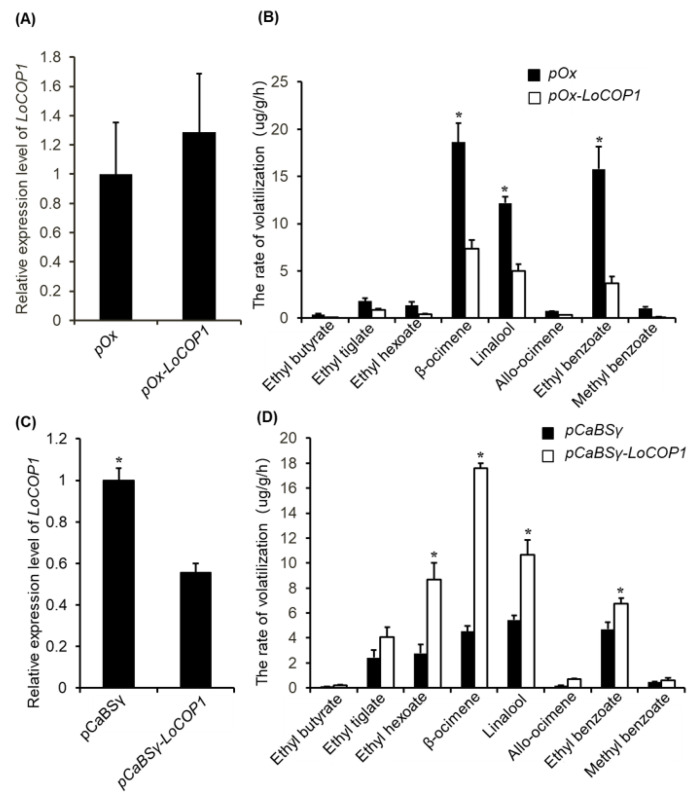
Gene transient overexpression and silencing alter volatile emission in *Lilium* ‘Siberia’ flowers. (**A**) *LoCOP1* transcript level in *Lilium* ‘Siberia’ around the injection sites after transient overexpression. (**B**) The rate of volatilization of the petal of *Lilium* ‘Siberia’. *LoCOP1* was used for overexpression (pOx-LoCOP1) with the pOx vector. Empty pOx-vector was used as control. (**C**) *LoCOP1* transcript level in *Lilium* ‘Siberia’ around the injection sites after VIGS. (**D**) The rate of volatilization of the petal of *Lilium* ‘Siberia’. LoCOP1 was used for suppression (BSMV-LoCOP1) with the γ-vector combined with α-vector and β-vector, Empty γ-vectors were used as controls. Asterisks indicate statistically differences (* *p* < 0.05). Error bars show the standard deviations from tree biological replicates.

**Figure 5 plants-12-02004-f005:**
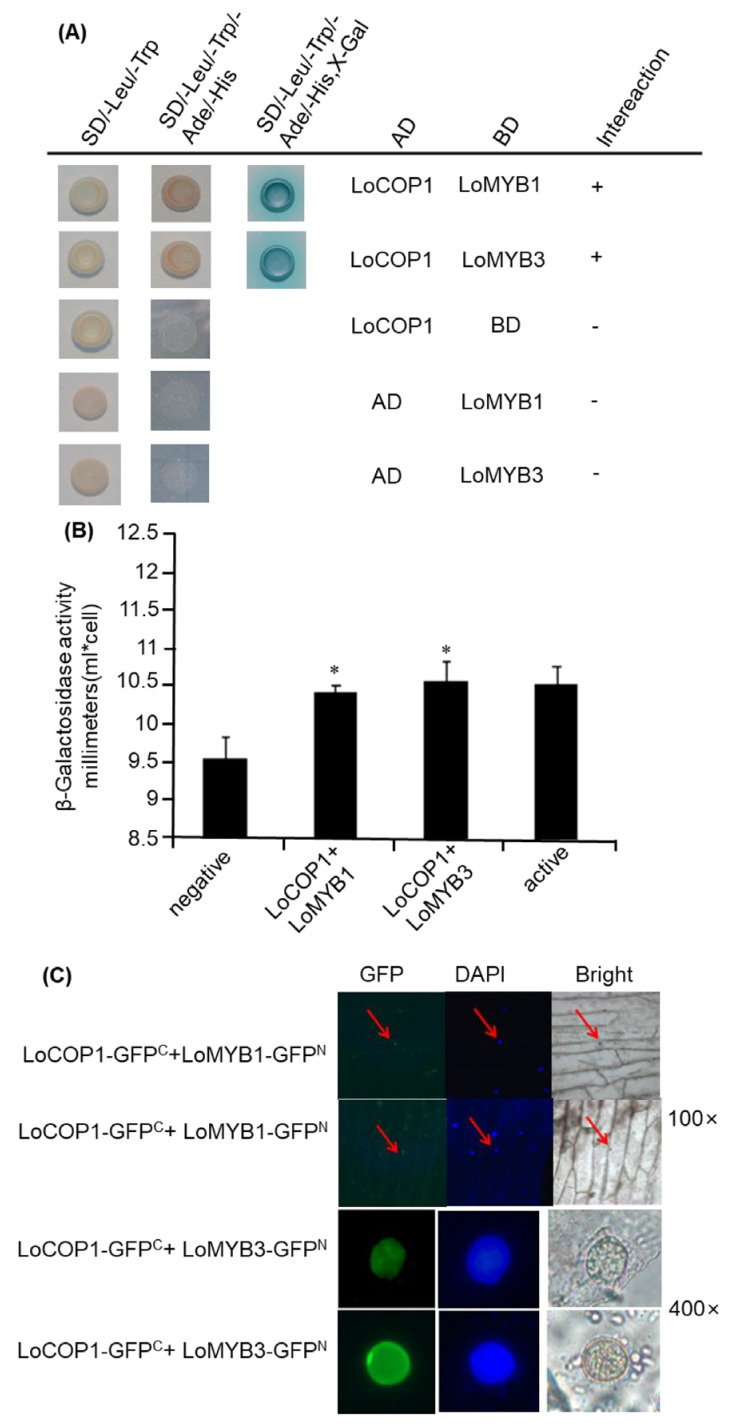
Interaction between LoCOP1 and LoMYB1 and LoMYB3. (**A**) Yeast two-hybrid assays were conducted with selective growth combined with a 5-bromo-4-chloro-3-indolyl-b-d-galactopyranoside acid overlay assay. (**B**) Result of β-galactosidase assays. (**C**) BiFC interaction assays using onion epidermal cells. GFP fluorescence was detected 2 days after transfection. Asterisks indicate statistically differences (* *p* < 0.05). Error bars show the standard deviations from tree biological replicates.

**Figure 6 plants-12-02004-f006:**
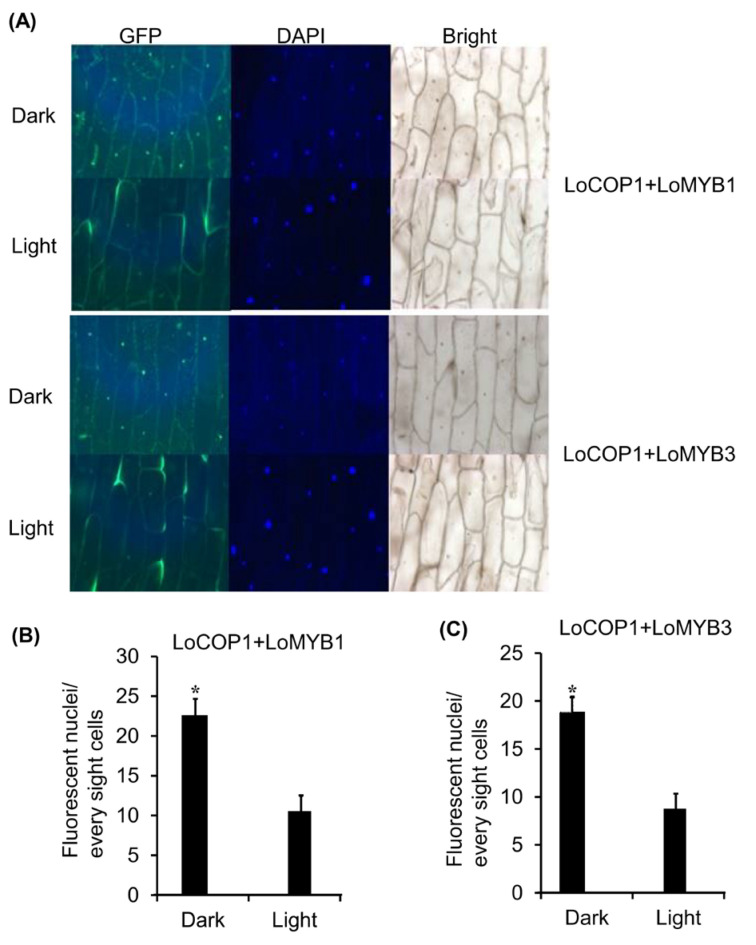
LoCOP1 interactions with LoMYB1 and LoMYB3 were controlled by light. (**A**) BiFC interaction assays using onion epidermal cells. Confocal microscopy of representative nuclei of BiFC interaction assays Pro35S: GFP^C^-LoCOP1 combined with Pro35S: GFP^N^-LoMYB1 and Pro35S: GFP^C^-LoCOP1 combined with Pro35S: GFP^N^-LoMYB3 in onion epidermal grown under light or continuous dark. (**B**) The numbers of fluorescent nuclei of the nuclei in onion epidermal cells expressing Pro35S: GFP^C^-LCOP1 combined with Pro35S: GFP^N^-LMYB1. (**C**) The numbers of fluorescent nuclei of the nuclei in onion epidermal cells expressing Pro35S: GFP^C^-LCOP1 combined with Pro35S: GFP^N^-LoMYB3 grown under white light or simulated shade conditions. GFP fluorescence was detected 2 days after transfection. Error bars show the standard deviations from tree biological replicates. Asterisks indicate statistically differences (* *p* < 0.05). Error bars show the standard deviations from tree biological replicates.

## Data Availability

All data were analyzed using LSD with Excel software. *p*-values < 0.05 were considered to be significant.

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
