# Peer review of "Light Regulation of LoCOP1 and Its Role in Floral Scent Biosynthesis in *Lilium* ‘Siberia’"

_plants, 2023, doi:10.3390/plants12102004_

Round 1

Reviewer 1 Report

The manuscript provides valuable information about regulatory mechanism in the light signaling system especialy mode of action of the constitutively photomorphogenic 1 (COP1) in Lilium 'Siberia.'. The study evaluates whether LoCOP1 affects the floral fragrance of Lillium 67 ‘Siberia’ by response to light changes.  Some minor writing errors were corrected (labeled and colored on pdf) within the text. This reviewer has the following concerns:

1. Before each use of an abbreviation, the full name must be used

2. As a lot of abbreviations are used in the article it is advisable to put them all at the beginning of the article with the expansions.

3. In the Discussion chapter - there is too much repetition of the results obtained. 

Author Response

Dear reviwer, please see the attachment.

Reviewer 2 Report

The authors investigated the function of LoCOP1 in fragrance production of Lilium ‘Siberia’ by overexpression and silencing of LoCOP1 in lily flowers. They found LoCOP1 interacts with the LoMYB1 and LoMYB3 proteins by yeast two-hybrid, β-galactosidase tests, and BiFC interaction assays. Overall, the findings should be of value for understanding volatile compound biosynthesis in lily flowers. However, there are several issues need to be addressed before publication.

The conclusion is that LoCOP1 interacts with LoMYB1 and LoMYB3 to regulate the release of floral fragrances, and such interaction is controlled by light. Since LoMYB1 and LoMYB3 are important for fragrance production which is also controlled by light, why did not investigate the relative expression of LoMYB1 and LoMYB3 between dark and light condition?

In yeast two-hybrid, does LoCOP1 have transcriptional self-activation activity?

Flowers at different development stages behave differently in the capacity of fragrance emission. Which stage of flowers was sampled in LoCOP1 overexpression and silencing for volatile collection?  

Line 14-15, “we isolated the COP1 gene from the petals of Lilium 'Siberia' (Lo- COP1) and investigated its function.” (Lo- COP1) should be placed after COP1 gene.  

Line33-34, add a more recent reference of floral scent: Mostafa et al. (2022) Front Plant Sci, 13:860157.

Line 49, “The emission of floral scent from Lilium ‘siberia’ in response to light intensity” The sentence is problematic in grammar.

Line 50-51, here is about the leaf monoterpenes. According to the context, here should be about flower fragrance.

Line 95-101, “tissue-specific and expression pattern analysis of LoCOP1 in Lilium ‘Siberia’ flower.”  However, some organs in the paragraph, such as the leaf, stem and root, are not flowers.

Line157-158, “We used yeast two-hybrid assay to evaluate whether LoCOP1 protein regulates key floral scent related LoMYB1 and LoMYB3 in lily or not.” How did you know that LoMYB1 and LoMYB3 are key floral scent related transcription factors in lily? How did you screen out these two TFs?

Line 214, “It to mediate” should be “It mediates”.

Line 228, change “relationship study” to “sequence alignment”.

Line 229, “a phylogenetic investigation” should be “our phylogenetic analysis”

Line 292, what does “floral pathway” mean? Pathway of volatile production?

Line337-340, LoMYB1, LoMYB2, and LoMYB3 function in producing fragrance compounds needs a citation.

Line353-369, “we cloned a gene called SUPPRESSOR OF PHYA-105 (SPA)……”. “Yeast results showed LoCOP1 can interact with LoSPA…..”. However, I did not find any results of LoSPA in the study.

Line 425-434, how did you overexpress LoCOP1 in Lilium flowers? Transient transformation by injection? Please provide the transformation details.

Author Response

(The authors gave the same response as above.)
